# No Evidence of Benefits of Host Nano-Carbon Materials for Practical Lithium Anode-Free Cells

**DOI:** 10.3390/nano12091413

**Published:** 2022-04-20

**Authors:** Bingxin Zhou, Baizeng Fang, Ivan Stoševski, Arman Bonakdarpour, David P. Wilkinson

**Affiliations:** Department of Chemical & Biological Engineering, The Clean Energy Research Center, University of British Columbia, 2360 East Mall, Vancouver, BC V6T 1Z3, Canada; bxzhou@mail.ubc.ca (B.Z.); bfang@chbe.ubc.ca (B.F.); ivan.stosevski@ubc.ca (I.S.); arman@chbe.ubc.ca (A.B.)

**Keywords:** nano-carbon based materials, lithium host, lithium anode-free cells, intercalation, deposition

## Abstract

Nano-carbon-based materials are widely reported as lithium host materials in lithium metal batteries (LMBs); however, researchers report contradictory claims as to where the lithium plating occurs. Herein, the use of pure hollow core-carbon spheres coated on Cu (PHCCSs@Cu) to study the lithium deposition behavior with respect to this type of structure in lithium anode-free cells is described. It is demonstrated that the lithium showed some initial and limited intercalation into the PHCCSs and then plated on the external carbon walls and the top surface of the carbon coating during the charging process. The unfavorable deposition of lithium inside the PHCCSs is discussed from the viewpoint of lithium-ion transport and lithium nucleation. The application potential of PHCCSs and the data from these LMB studies are also discussed.

## 1. Introduction

Li-ion batteries and Li-metal batteries have been extensively investigated for use in diverse applications [1,2,3,4]. Various nano-structured carbon materials, such as hollow nanospheres, nanofibers, and nanotubes, have been reported as lithium anode hosts to regulate lithium deposition behavior in lithium metal batteries (LMBs) [5,6,7,8,9]. Cui et al. claimed that lithium ions can penetrate hollow-nanosphere-layered films and layered reduced-graphene oxide films and deposit underneath them as metallic lithium [10,11]. Liu et al. also claimed that lithium first deposits inside carbon hollow fibers, followed by plating on outer surfaces [12]. However, a later publication demonstrated that lithium preferentially deposits outside carbon walls when there are no lithiophilic nucleation sites inside the spheres [8]. Cui et al. also claimed, in another paper, that the tortuosity of the carbon host has a significant effect on the lithium deposition preference [13]. Moreover, Ye et al. proposed that lithium tends to deposit inside carbon shells only when there are enough lithiophilic functional groups and when the specific surface area (SSA) of the carbon spheres is high enough [14]. Therefore, the current literature on lithium deposition in carbonaceous hosts includes contradictory conclusions. Despite a large effort toward the development of carbon host materials, with claims that Li plates inside various carbon cavities or hollow structures and improves the cycle life, a simple consideration of the overpotential and the resistive losses present implies that lithium preferentially deposits at the sites with the shortest diffusion paths and the lowest nucleation overpotential [8,15,16]. In this case, the deposition of Li on the outer sides of the host carbon shell is more likely as it results in shorter diffusion pathways. Therefore, it seems, with these contradictory results, that further investigations are critical to elucidate the nature of lithium deposition with respect to pure carbon hosts.

For this study, we synthesized pure hollow core-carbon spheres (PHCCSs) and used them as a lithium host to investigate the behavior of lithium plating. We demonstrate that in the absence of lithophilic functional groups inside the core, lithium ions intercalate within the carbon first and then plate outside the PHCCSs during the charge reaction. The possibility of lithium depositing inside the PHCCSs according to the characteristics of the carbon materials and lithium ions is discussed in detail. In addition, the theoretical volumetric capacity of the PHCCS-based anode electrode is calculated, and suggestions on carbon size are proposed in order to outperform the currently used graphite anode. Finally, we show that the demonstration of high and stable columbic efficiency (CE), which has become customary in the field, does not necessarily imply stable capacity retention in LMBs, and the cycling performance of volumetric (and specific) capacities continue to be critical metrics that should be reported.

## 2. Materials and Methods

Preparation of PHCCSs@Cu. The synthesis method of PHCCSs has been reported in detail elsewhere [17,18]. The electronic conductivity of PHCCSs is about 1.0 S/cm [19]. Three kinds of PHCCS, with different core sizes and shell thicknesses (core diameter: shell thickness = 110 nm: 60 nm; 270 nm: 55 nm; 360 nm: 55 nm) were synthesized. The PHCCS coating slurry consisted of PHCCS: SBR (styrene-butadiene rubber): CMC (carboxymethyl cellulose) = 84: 8: 8 (wt.%), and the resulting PHCCS coating thickness was ≈15 µm, with a loading of 0.9 mg/cm^2^. The schematics of the synthesis procedure, scanning electron microscopy (SEM) images, X-ray diffraction analysis (XRD), thermogravimetric analysis (TGA) plots, and the structural parameters from N_2_ adsorption–desorption isotherms (P/P_0_ ranges from 0.01 to 1.0) are shown in Appendix A.

Structure Characterizations. XRD patterns of PHCCSs were obtained with an X-ray powder diffractometer (D2 Phaser, Bruker AXS GmbH. Kalsruhe, Germany) in the 2*θ* range of 5–80° in steps of 0.02^o^ using Cu Kα radiation as the X-ray source and beam characteristics of 35 kV and 40 mA. TGA analyses were conducted with a thermal analyzer (Shimadzu TG-50H) in the temperature range of 15–900 °C under an air flow at 100 mL/min. N_2_ adsorption and desorption isotherms were measured at 77 K on a Micromeritics ASAP-2020 gas adsorption analyzer (Micromeritics Instrument Corporation, Norcross, GA, USA) after a sample was degassed at 423 K to 20 mTorr for 12 h. The specific surface areas were determined from nitrogen adsorption using the Brunauer−Emmett−Teller (BET) method. Total pore volumes (V*_Total_*) were determined from the amount of gas adsorbed at a relative pressure of 0.99. Micropore volume (V*_Micro_*) and micropore size of the porous materials were calculated from the analysis of the adsorption isotherms using the Horvath−Kawazoe (HK) method. Pore size distribution (PSD) was calculated from the adsorption branches by the Barrett−Joyner−Halenda (BJH) method. SEM (FEI, Helios NanoLab 650, FEI Company, Hillsboro, OR, USA) imaging was carried out to visualize the morphology of PHCCSs and disassembled cell electrodes. Transmission electron microscopy (TEM) analyses were conducted with an FEI Tecnai G2 microscope (FEI Company, Hillsboro, OR, USA) operated at 200 kV. Coin cells after cycling were disassembled for characterization by ex situ SEM. Before testing, the disassembled electrodes were rinsed with DME solvent to remove residual electrolytes and then dried in an Ar filled glovebox (O_2_ and H_2_O < 1 ppm, LabStar, MBraun Inc., Stratham, NH, USA). Energy-dispersive spectroscopy (EDS, EDAX Octane Super 60 mm^2^ EDX detector) and elemental analysis were employed to detect the elemental compositions of the obtained samples.

Electrochemical Measurements. Electrochemical measurements were performed in coin cell hardware (CR2032, MTI Corporation) using a Biologic potentiostat (VMP3 Multichannel Potentiostat, Biologic Science Instruments, Seyssinet-Pariset, France). The working electrode was copper foil (≈27 µm, 99.9%, basic copper) or PHCCSs@Cu (coating thickness of ≈15 µm, 0.9 mg/cm^2^) while the counter electrode was lithium foil (99.9%, Sigma-Aldrich, St. Louis, MO, USA) or LiCoO_2_ (~55 µm, 20–22.5 mg/cm^2^, MTI Corporation, Richmond, CA, USA). One layer of Celgard separator (16 µm) was used to separate the electrodes. The electrolyte consisted of 2M lithium bis(fluorosulfonyl)imide (LiFSI, 99.9%, Solvionic, Toulouse, France) and 1M lithium bis(trifluoromethanesulfonyl)imide (LiTFSI, 99.95%, Sigma-Aldrich, St. Louis, MO, USA) in 1:1 (*v*/*v*) 1,3-dioxolane (DOL, 99.5% Sigma-Aldrich)/1,2-dimethoxyethane (DME, 99.5%, Sigma-Aldrich); 50 µL of electrolyte was added to each cell. This type of electrolyte was selected because it produces a relatively stable SEI, which can efficiently suppress Li dendrite formation and enhance the coulombic efficiency and is, therefore, widely used in LMB experiments [20]. After assembling and resting for 24 h, the cells were cycled with galvanostatic charge/discharge at 25 °C. For the cells with Li as the counter electrode, the electrochemical cycling consisted of (a) charge at C/5 for 5 h (voltage limit of 2 V), and (b) discharge at C/2 for 2 h (voltage limit of −0.5 V vs. Li/Li^+^); for the cells with LiCoO_2_ as the counter electrode, the electrochemical cycling consisted of (a) charge at C/5 for 5 h (voltage limit of 4.2 V), and (b) discharge at C/2 for 2 h (voltage limit of 2.5 V). The charging capacity limitation was set as 3.13 mAh unless otherwise specified. For each experiment, at least two cells were tested to assure repeatability.

Electrochemical measurements were also performed on a homemade optical glass cell (Figure 1). Bare copper rods (purity > 99%, diameter of 3.2 mm, McMaster-Carr, Santa Fe Springs, CA, USA) or coated with PHCCS were used as the working electrode, and a lithium rod (>99.9%, diameter of 3.2 mm, Sigma-Aldrich, St. Louis, MO, USA) was used as the reference and counter electrode. The optical cells were assembled and sealed in the glovebox. Imaging was performed with an optical microscope (XTL-2400, OKA, Wuzhou, China) and electrochemical measurements were conducted by a Biologic potentiostat. In situ images and videos were recorded by a digital camera (A3530U, OMAX, Kent, WA, USA) and analyzed with Toupview software (version: x64, 4.11.17852.20201017, Hangzhou ToupTek Photonics Co., Ltd., Hangzhou, China).

## 3. Results and Discussion

Assuming that the lithium could pass through the mesoporous carbon shell and deposit inside the hollow core space, and assuming the complete utilization of the inner volume of the PHCCSs’ hollow space for the dense metallic plating of the lithium, it was possible to calculate the theoretical capacity using the core dimensions, the shell thickness, the PHCCS loading on the copper foil, and the spherical symmetry of the carbon particles (detailed calculations are provided in Appendix A). The calculated theoretical capacity of the PHCCSs@Cu anode (core diameter: shell thickness = 110 nm: 60 nm, thickness of ~15 µm) based on the complete filling of the core with Li was 0.372 mAh. The PHCCSs@Cu were then assembled into coin cells as the anode and tested with different capacity limitations according to the calculated theoretical capacity data. The cells were disassembled after the fourth charging step and the PHCCSs@Cu anodes were then examined by SEM. As shown in Figure 2(a1,b1),when the charging capacity was set to the hollow core (geometrical) capacity of 0.372 mAh, no lithium deposition was detected in either the top-view or the cross-sectional SEM images. Only two layers, the carbon-coating layer and the Cu layer, were observed from the cross-sectional image. On the other hand, when the capacity limitation was set as 3.13 mAh (full cathode capacity, ≈9× of the hollow core capacity), SEM images clearly show the presence of a thick (≈20 μm) lithium layer on the top surface of the PHCCSs coating (Figure 2(a2,b2)) in addition to the carbon-coating layer and Cu layer (the copper layer was partially covered by the carbon coating owing to the cutting process of the electrode). It is worth noting that overall, the thickness of the carbon, which was measured by a micrometer in different locations, was similar; the thickness of the carbon in Figure 2 shows minor variations because of the location chosen. The chunky lithium morphology observed in this study is typical of lithium in LIFSI-LITFSI (DME/DOL)-based electrolytes, as reported in [21].

Figure 2(c1,c2) show the magnified images of the lithium deposition in the carbon layer. No significant structural changes (or collapsing) of the PHCCSs can be observed, clearly showing that the PHCCSs maintained their original structure. We observed that carbon spheres remained visible and intact with the lower capacity (0.372 mAh, shown in Figure 2(c1)), while the carbon particles and their surroundings became covered by petal-like Li particles (Figure 2(c2)) at the higher capacity (3.13 mAh). As a similar structure was observed in a previous study [22]. This comparison clearly indicates that the lithium deposited on the outer surfaces and in the interstitial gaps of the carbon particles. Overall, no evidence for lithium deposition inside the PHCCS was observed through the SEM analysis.

To provide further insight into the deposition mechanism, the voltage–capacity curves of the Li|PHCCSs@Cu cell were carefully examined. Figure 3a shows that when the charging capacity was limited to the geometrical capacity of the hollow core volume, the cell potential remained above 0 V during the entire charging process, implying that some lithium intercalation into carbon occurred. Figure 3b shows that a charging capacity of ≈0.5 mAh was consumed when the cell potential remained above 0 V, further supporting the occurrence of the intercalation phenomenon. This observation also explains the deposition of a lithium layer when the capacity was set to 3.13 mAh, and the lack of a lithium layer when the capacity was limited to 0.372 mAh, well below the intercalation capacity of ≈0.5 mAh. In fact, Dahn et al. have demonstrated that lithium ions can insert reversibly into most carbonaceous materials [23]. Ye et al. also demonstrated that lithium ions intercalate into carbon shells before deposition, even if the carbon shell is amorphous [14]. In addition, it is widely reported that the characteristic shape of the voltage curves for the intercalation of lithium into disordered carbon is not that of a flat and low potential plateau (as in graphite), but rather that of a gradually decreasing curve, which corresponds to the case of our non-graphitized carbon samples, further suggesting that some lithium intercalation occurred before the metallic deposition of lithium. When the charging C-rate increased to 5 C (Figure 3c), the more severely induced polarizations hindered the intercalation process, and the lithium ions tended to deposit as metallic lithium directly, without intercalation. The discharge capacity of the first cycle, which was much lower than that of the second and third cycle, can be attributed to the SEI formation and other passivation processes, including the possibility that the inserted Li-ions were trapped in the inserted sites and also led to irreversible capacity [24]. Figure 3d shows the long-term cycling performance of the PHCCSs@Cu|LiCoO_2_. The core sizes and shell thicknesses of the PHCCSs did not significantly affect the capacity fade and CE performance, further confirming that no significant (if any) lithium deposition occurred in the core. Because of the higher porosity and larger specific surface area of the PHCCSs compared with bare Cu, the deposited lithium had greater exposure to the electrolyte, and was thus consumed at a higher rate during the cycling process, leading to the inferior cycling performance of the PHCCSs@Cu|LiCoO_2_ cells compared to that of the Cu|LiCoO_2_ cell.

Besides the lithium deposition behavior of the PHCCSs, we would like to emphasize a subtle but critical point, with respect to the presentation and analysis of the data in this research area and in the more general field of LMB research. It has become customary to report the stability of cycling life of lithium (or “lithium free-anode”) using the CE metric in battery research as it reflects the Li-ion loss of individual cycles during the cycling process [25]. A significant number of published papers on LMBs show relatively stable CE and claim the excellent performance of their cells. However, having a “high” and stable CE does not necessarily imply high-capacity retention in LMBs [26,27]. We show here, in Figure 3d, that for PHCCSs, the CE remains high and relatively stable during the cycling process, but the specific capacity fades quickly. It is therefore insufficient, from a data presentation point of view, to only report the CE vs. cycle number. The cycling performance of full-cell-capacity data (specific capacity, volumetric capacity, capacity retention, etc.) must also be reported in research papers to avoid misleading conclusions. It is also worth noting that the CE of PHCCSs@Cu was even lower than that of the Cu during the initial cycles, which was mainly because PHCCSs are not perfectly layered structures, as in the case of graphite; hence, the reversibility of intercalated lithium ions is smaller than that of fully graphitized carbon materials.

Figure 4 shows the in situ images of the deposition of the lithium on the PHCCSs@Cu, along with the voltage–capacity curves. The current densities were set to 0.5, 1, and 2 C-rates in the coin cell (or 1.565, 3.13, and 6.26 mA/cm^2^, respectively). The voltage of the PHCCSs@Cu|Li cells dropped below 0 V after applying the current for 60 min, 24 min, and 10 min (blue, red, and green lines), respectively, under these three C-rates. By contrast, the voltage of the Cu|Li cell dropped below 0 V in less than 1 min when applying the current (black line). The gradual voltage changes in the PHCCSs@Cu|Li cells implies the lithium’s intercalation into the carbon coating, which is consistent with the coin-cell results. On the other hand, high current densities led to reduced capacities of the lithium intercalation, which was also consistent with the coin-cell results. Furthermore, the images recorded under 1 C during the lithium deposition process show the appearance of lithium at the copper surface and the carbon-coated surface at the same time, indicating that lithium prefers to deposit at any location that has a low nucleation overpotential and a short diffusion pathway, further demonstrating the difficulty of the lithium passing through the carbon walls and depositing inside the carbon spheres. A complete video of the lithium intercalation/deposition process was recorded and is provided in Appendix A.

We now discuss the possibility of lithium deposition inside the hollow core space considering the schematic shown in Figure 5a. Based on the results shown thus far, we know Li-ions intercalate into carbon shells first, followed by plating, which is similar to the results presented by Ye et al. [14]. If lithium is to plate inside the hollow cores of carbon spheres, lithium ions need to pass through the carbon shell and then deposit inside. Lithium, however, tends to deposit at the sites that show the shortest diffusion pathways and have the lowest nucleation overpotential [8,15,16]. This means that lithium prefers to deposit on the external walls of carbon shells. If there are no lithiophilic functional groups inside the carbon spheres that can provide sites with lower nucleation overpotential and higher binding energy, there is no driving force for lithium ions to pass through the intercalated carbon shell. Thus, the passage of lithium ions through pure carbon shells and their deposition inside is highly questionable.

On the other hand, based on the pore size of the as-prepared PHCCS (average diameter of ≈3.8–4.1 nm, see Appendix A for size distribution) and the typical length scales of solvated Li^+^ with different anions (≈1–2 nm) reported in the literature [28,29,30,31,32], theoretically, solvated lithium ions can pass directly through pores. In addition, there were cracks and defects on the carbon shell, which may have provided conduits for the lithium ions to pass through. However, according to Wang et al. [33], the current density within the pores and cracks should be larger than that at the inner surface of the carbon shell because of the higher curvature; hence, the lithium should preferentially deposit inside the pores in the vicinity of the external surface. This means that lithium ions gradually fill up these transport channels and prevent the passage of additional Li-ions to deposit inside. Moreover, even if lithium ions could penetrate inside, their quantity would be very limited, and the imposed resistance would increase compared to the direct plating of Li on the outer surfaces of the carbon shell.

In summary, based on these observations, the likelihood of lithium plating in the internal void volumes of hollow carbon structures is very low. The preferential deposition of lithium on outer surfaces is irrespective of the type and nature of porous carbonaceous (or non-carbonaceous) structures. However, chemical modification, such as through the incorporation of lithophilic groups inside the hollow space, may have some impact, since it could change the driving force of the inner-surface plating of lithium to some extent [8,14]. Nevertheless, the kind of chemical modification, which may induce significant changes in the physics of lithium deposition and, consequently improve the battery performance, requires more careful investigation.

Even if there is a way to cause lithium to preferentially and reversibly deposit inside hollow core volumes, it is still necessary to ensure that the volumetric capacity of the lithium host coating remains higher than that of the standard graphite anode, and that this capacity is maintained over many cycles, when designing a porous host structure. It is only in this case that the lithium host approach could show any practical advantage over the currently used graphite anode materials. We calculated the theoretical volumetric capacity of the PHCCS anode coatings with different core sizes (diameters ≈ 110–360 nm) and shell thicknesses (≈30–60 nm), and an active material content of 97% (example capacity calculation parameters are shown in Appendix A). As shown in Figure 5b, only PHCCSs with capacities higher than 400 mAh/cm^3^ (i.e., graphite’s volumetric capacity [34]) should be considered as lithium hosts for use in anode electrodes. This type of calculation should also be applied in all lithium-host designs, regardless of the materials or structures that are used. Besides the volumetric capacity, the fabrication process and cost, as well as the environmental impact, are also of great significance in terms of commercial application and should be carefully considered during the design.

## 4. Conclusions

We have shown that with PHCCSs, lithium ions intercalate into carbon and then plate outside PHCCSs during the charging step. No metallic lithium deposition inside the hollow carbon spheres was observed, i.e., there is no evidence for the benefits of host nano-carbon materials for practical lithium anode-free cells. Using the physical dimensions of PHCCSs and lithium ions, the possibility of lithium depositing inside the hollow space was examined, further demonstrating the difficulty of lithium depositing inside pure carbon hosts. Although the PHCCSs were utilized here as a model system to demonstrate the lithium deposition behavior, the analysis also applies to nano-structural carbon hosts of other shapes. On the other hand, even if the reversible plating of lithium in internal pores can be achieved with full utilization, porous host dimensions (internal diameter and shell thickness) need to be carefully considered in order to outperform the conventional intercalation of graphite-based anode electrodes. Moreover, we have shown that a high and stable CE does not necessarily imply stable capacity retention in lithium metal batteries, and rigorous data analyses must always provide information as to the volumetric/specific capacity vs. the cycle number.

## Figures and Tables

**Figure 1 nanomaterials-12-01413-f001:**
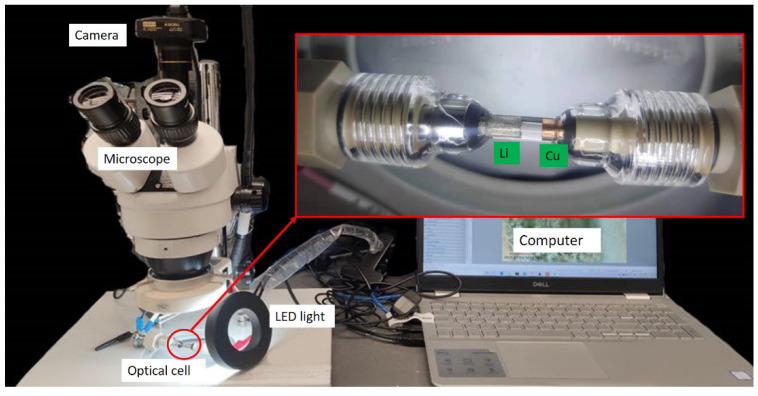
The setup of the optical cell test.

**Figure 2 nanomaterials-12-01413-f002:**
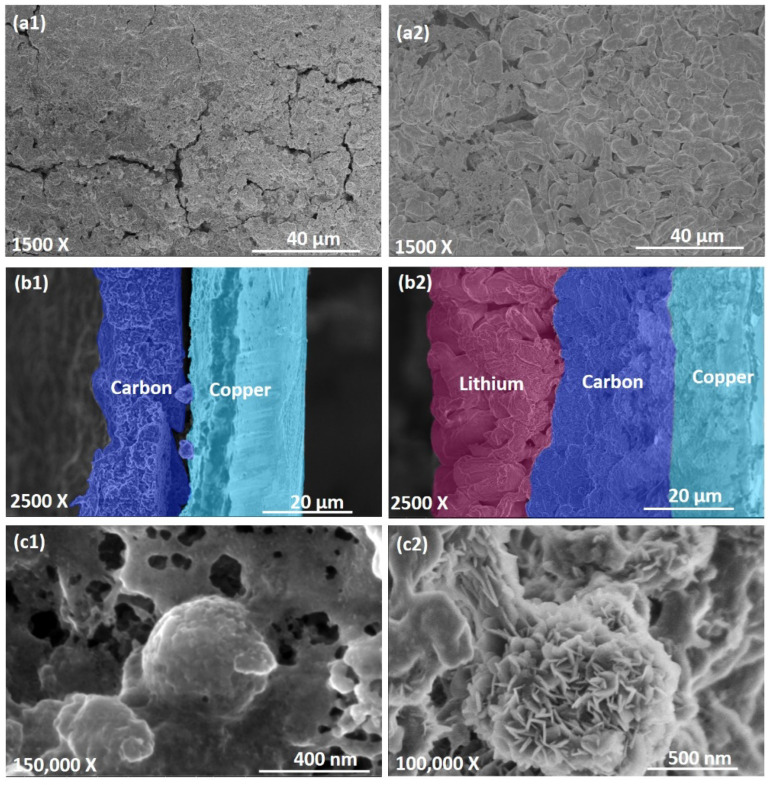
SEM images and X-ray maps of PHCCSs@Cu anode (core diameter: shell thickness = 110 nm: 60 nm) in the charged state after the fourth cycle, capacity limitations of 0.372 mAh (**a1**–**c1**) and 3.13 mAh (**a2**–**c2**), respectively. Top-view images (**a1**,**a2**), cross-sectional images (**b1**,**b2**), and magnified images of carbon particles (**c1**,**c2**).

**Figure 3 nanomaterials-12-01413-f003:**
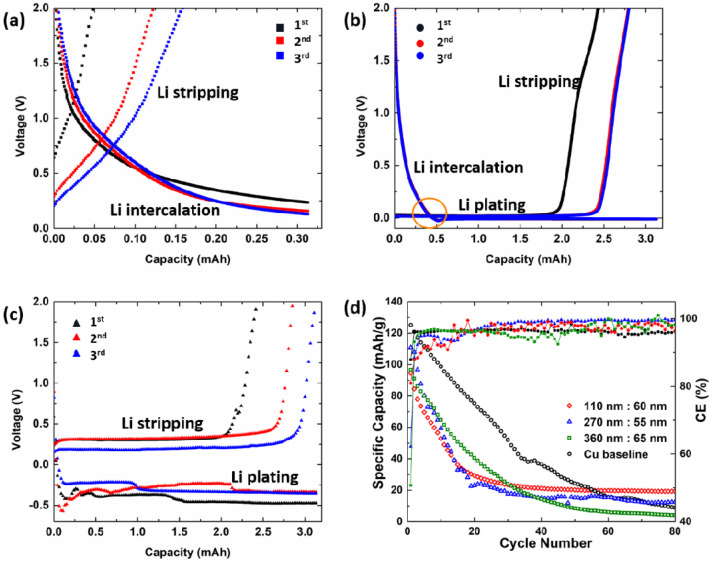
Voltage–capacity plots of the first, second, and third cycle of Li|PHCCSs@Cu cells recorded under different capacity limitations: (**a**) 0.372 mAh (geometric hollow core capacity, charging/discharging C-rates of 0.2 C/0.5 C (0.626 mA/1.565 mA)), (**b**) 3.13 mAh (full cathode capacity, ≈9× of geometric hollow core capacity, charging/discharging C-rates of 0.2 C/0.5 C (0.626 mA/1.565 mA)) and (**c**) 3.13 mAh (charging/discharging C-rates of 5 C/5 C (15.65 mA/15.65 mA)); (**d**) specific capacity and CE of Cu|LiCoO_2_ and various sizes of PHCCSs@Cu|LiCoO_2_ full cells (charging/discharging C-rates of 0.2 C/0.5 C (0.626 mA/1.565 mA)).

**Figure 4 nanomaterials-12-01413-f004:**
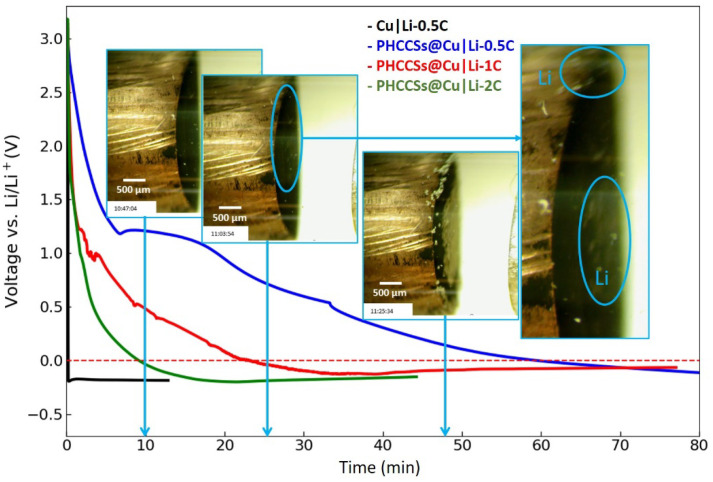
The voltage vs. time profile of Cu|Li and PHCCSs@Cu|Li. The images provided show the deposition of lithium in PHCCSs@Cu|Li optical cell during the lithium deposition process (c-rate of 1C; the blue circles show the locations where plated lithium first appears).

**Figure 5 nanomaterials-12-01413-f005:**
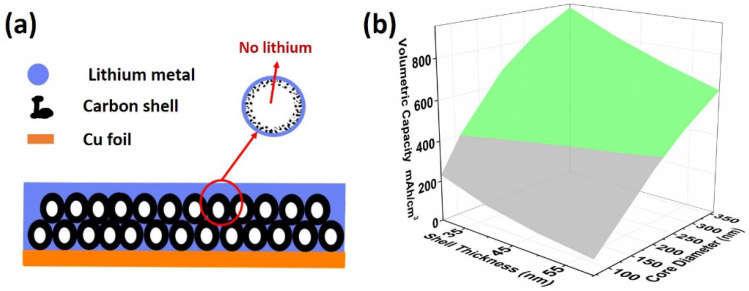
(**a**) Schematic of the mechanism of lithium deposition inside PHCCSs, and (**b**) theoretical volumetric capacity of PHCCS coatings of different core sizes and shell thicknesses.

## Data Availability

Data are contained within the article or Appendix A.

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
