# Peer review of "No Evidence of Benefits of Host Nano-Carbon Materials for Practical Lithium Anode-Free Cells"

_nanomaterials, 2022, doi:10.3390/nano12091413_

Round 1

Reviewer 1 Report

In this manuscript, the authors investigate the lithium deposition behavior in the lithium anode-free cell by using hollow core-carbon spheres (PHCCSs). They have shown that the lithium depositing inside the hollow carbon is unfavorable, and the host nano-carbon materials is not beneficial for practical lithium anode-free cells. Additionally, this analysis can be also applied to other nano-structural carbon hosts with different shapes. After carefully reviewing the manuscript, minor revisions are still needed for the publication in the Nanomaterials.

  1. HR-TEM images are suggested to be applied to better characterize the micromorphology of the synthesized hollow core-carbon spheres.
  2. The thickness of hollow core-carbon spheres seems quite different in the Fig. 2(b1, b2), and it could be better to explain the effect of thickness on deposition behavior.
  3. The caption denoted with “1, 2, 3” in the Fig. 3 should be changed to “1st, 2nd, 3rd,”.
  4. The initial irreversible capacity is an important parameter to evaluate the performance of anode materials. In the article, the initial Coulombic efficiency of PHCCSs@Cu was lower than that of pure Cu, what role did PHCCSs play in the host?
  5. In the Fig. 3d, it would be better to re-draw the dot types as it is difficult to distinguish the capacity and Coulombic efficiency.

Author Response

In this manuscript, the authors investigate the lithium deposition behavior in the lithium anode-free cell by using hollow core-carbon spheres (PHCCSs). They have shown that the lithium depositing inside the hollow carbon is unfavorable, and the host nano-carbon materials is not beneficial for practical lithium anode-free cells. Additionally, this analysis can be also applied to other nano-structural carbon hosts with different shapes. After carefully reviewing the manuscript, minor revisions are still needed for the publication in the Nanomaterials.

Reply:  We would like to thank the reviewer for the positive assessment and the constructive comments. We are now carefully addressing all the points raised and hope the revised manuscript is acceptable for publication.

  1. HR-TEM images are suggested to be applied to better characterize the micromorphology of the synthesized hollow core-carbon spheres.

Reply: We would like to thank the reviewer for the comment. As you suggested, TEM images of the hollow core-carbon spheres are now provided (as shown in Fig. S1c, Fig. S1d) in order to better characterize their micro-morphologies. These images allow us to estimate the core diameter and shell thickness more easily. These values are in excellent agreement with those obtained from their SEM images. 

  1. The thickness of hollow core-carbon spheres seems quite different in the Fig. 2(b1, b2), and it could be better to explain the effect of thickness on deposition behavior.

Reply: We would like to thank the reviewer for the comment. The coating thickness of PHCCSs is set to 15 µm during coating in this study (please see the main text, line 66), the thickness was kept almost constant for the PHCCSs of different core sizes and shell thicknesses. Overall the thickness of carbon is the same, it is measured by micrometer at different locations, and the thickness of carbon in the figure here shows minor variations because of the location chosen. That is now included in the manuscript (line 145-147).

  1. The caption denoted with “1, 2, 3” in the Fig. 3 should be changed to “1st, 2nd, 3rd,”.

Reply: We would like to thank the reviewer for the comment. The caption has been changed now as suggested.

  1. The initial irreversible capacity is an important parameter to evaluate the performance of anode materials. In the article, the initial Coulombic efficiency of PHCCSs@Cu was lower than that of pure Cu, what role did PHCCSs play in the host?

Reply: We would like to thank the reviewer for the comment. The PHCCSs do not have a perfect layered structure like graphite, and the lithium ions intercalated into the PHCCSs could not be fully extracted during the lithium stripping, hence a lower Columbic efficiency, especially during the initial cycles. The explanation has also been added to the main text (line 207-211).

  1. In the Fig. 3d, it would be better to re-draw the dot types as it is difficult to distinguish the capacity and Coulombic efficiency.

Reply: We would like to thank the reviewer for the comment. Fig. 3d has been revised as suggested.

Reviewer 2 Report

In the work, the authors studied the lithium deposition behavior with respect to the type of structure in lithium anode-free cells. The unfavorable deposition of lithium inside the pure hollow core-carbon spheres is discussed from the viewpoint of lithium-ion transport and lithium nucleation. This topic is very interesting by the community of the energy storage field and the strategy introduced in this study is well organized. The manuscript content is a quality fit for publication in this journal as a current form. However, the manuscript should be blushed up in English. The authors had better re-write the manuscript grammatically.

Author Response

In the work, the authors studied the lithium deposition behavior with respect to the type of structure in lithium anode-free cells. The unfavorable deposition of lithium inside the pure hollow core-carbon spheres is discussed from the viewpoint of lithium-ion transport and lithium nucleation. This topic is very interesting by the community of the energy storage field and the strategy introduced in this study is well organized. The manuscript content is a quality fit for publication in this journal as a current form. However, the manuscript should be blushed up in English. The authors had better re-write the manuscript grammatically.

Reply: We would like to thank the reviewer for the comment. We have carefully checked the manuscript and improved the English. We hope our changes have rendered the manuscript ready for publication.

Reviewer 3 Report

The manuscript entitled “No Evidence of Benefits of Host Nano-carbon Materials for Practical Lithium Anode-free Cells” by Bingxin Zhou et al. focuses on the study of the lithium deposition behavior with respect to the Pure Hollow Core-Carbon Spheres coated on Cu (PHCCSs@Cu) in lithium anode-free cells.

The authors demonstrated that lithium shows some initial and limited intercalation into the PHCCSs and then plates on the external carbon walls and the top surface of the carbon coating during the charging process. Furthermore, the authors discussed the unfavorable deposition of lithium inside the PHCCSs from the viewpoint of lithium-ion transport and lithium nucleation and the application potential of PHCCSs in Lithium Metal Batteries (LMB).

The manuscript is interesting and suitable to be published in this journal with minor revisions, which are summarized in the following points:

  • the introduction does not consider some articles published, that deal experimental and theoretical studies on the batteries advanced applications, so it has to be expanded;
  • the general form of the manuscript can be improved.

In order to improve the scientific quality of the manuscript, the reviewer's suggestions, corrections and questions follow:

  1. Introduction

In order to broad readership on the advanced batteries’ applications, the reviewer does recommend the authors to include the following relevant articles on the advanced applications of batteries to improve the reference part:

Advanced applications of batteries:

  1. International Journal of Hydrogen Energy, 2014, 39(24), pages: 12934-12947, DOI: 10.1016/j.ijhydene.2014.05.135
  2. Fuel Cells, 2016, 16(5), pages: 628-639, DOI: 10.1002/fuce.201500174
  3. Cogent Engineering, 2017, 4: 1357891, DOI: 10.1080/23311916.2017.1357891
  4. International Journal of Hydrogen Energy, 2017, 42(5), pages: 3166-3184, DOI: 1016/j.ijhydene.2016.12.082
  5. Journal of Energy Storage, 2016, 8, pages: 235-243, DOI: 10.1016/j.est.2016.08.012
  6. International Journal of Hydrogen Energy, 2019, 44(45), pages: 24895-24904, DOI: 10.1016/j.ijhydene.2018.12.038
  7. International Journal of Hydrogen Energy, 2019, 44(16), pages: 8479-8492, DOI: 10.1016/j.ijhydene.2019.02.003
  8. Energy Conversion and Management, 2020, 204, Article 112319, DOI: 10.1016/j.enconman.2019.112319.

  1. Materials and Methods

Line 63: the authors have to define the acronyms “SBR” and “CMC” the first time they are used.

Line 68: the authors have to insert the unit of relative pressure.

Line 104: the authors have to clarify the meaning of negative voltage.   

  1. Results and discussion

Figure 3: The sizes of graphics have to increase for improving the reader understanding.

Author Response

The manuscript entitled “No Evidence of Benefits of Host Nano-carbon Materials for Practical Lithium Anode-free Cells” by Bingxin Zhou et al. focuses on the study of the lithium deposition behavior with respect to the Pure Hollow Core-Carbon Spheres coated on Cu (PHCCSs@Cu) in lithium anode-free cells.

The authors demonstrated that lithium shows some initial and limited intercalation into the PHCCSs and then plates on the external carbon walls and the top surface of the carbon coating during the charging process. Furthermore, the authors discussed the unfavorable deposition of lithium inside the PHCCSs from the viewpoint of lithium-ion transport and lithium nucleation and the application potential of PHCCSs in Lithium Metal Batteries (LMB).

The manuscript is interesting and suitable to be published in this journal with minor revisions, which are summarized in the following points:

  • the introduction does not consider some articles published, that deal experimental and theoretical studies on the batteries advanced applications, so it has to be expanded;
  • the general form of the manuscript can be improved.

In order to improve the scientific quality of the manuscript, the reviewer's suggestions, corrections and questions follow:

Reply: We would like to thank the reviewer for these comments. Based on these comments (and other reviewer’s comment), we have further improved the quality of the manuscript. We believe the revised manuscript is now acceptable for publication.

Comment 1: Introduction

In order to broad readership on the advanced batteries’ applications, the reviewer does recommend the authors to include the following relevant articles on the advanced applications of batteries to improve the reference part:

Advanced applications of batteries:

  1. International Journal of Hydrogen Energy, 2014, 39(24), pages: 12934-12947, DOI: 10.1016/j.ijhydene.2014.05.135
  2. Fuel Cells, 2016, 16(5), pages: 628-639, DOI: 10.1002/fuce.201500174
  3. Cogent Engineering, 2017, 4: 1357891, DOI: 10.1080/23311916.2017.1357891
  4. International Journal of Hydrogen Energy, 2017, 42(5), pages: 3166-3184, DOI: 1016/j.ijhydene.2016.12.082
  5. Journal of Energy Storage, 2016, 8, pages: 235-243, DOI: 10.1016/j.est.2016.08.012
  6. International Journal of Hydrogen Energy, 2019, 44(45), pages: 24895-24904, DOI: 10.1016/j.ijhydene.2018.12.038
  7. International Journal of Hydrogen Energy, 2019, 44(16), pages: 8479-8492, DOI: 10.1016/j.ijhydene.2019.02.003
  8. Energy Conversion and Management, 2020, 204, Article 112319, DOI: 10.1016/j.enconman.2019.112319.

    Reply: We would like to thank the reviewer for the comment. As suggested, some relevant articles on the advanced applications of batteries particularly Li batteries have been added in the revised manuscript (line 23-24), for example, a) reference 5 (F. Sergi et al, Characterization and comparison between lithium iron phosphate and lithium-polymers batteries, Journal of Energy Storage, 2016, 8, 235-243); b) reference 8 (Jiangyun Zhang et al, Characterization and experimental investigation of aluminum nitride-based composite phase change materials for battery thermal management, Energy Conversion and Management, 2020, 204, 112319); c) Paul Albertus et al, Challenges for and pathways toward Li-metal-based all-solid-state batteries, ACS Energy Lett. 2021, 6, 1399–1404; d) Clare P. Grey and David S. Hall, Prospects for lithium-ion batteries and beyond—a 2030 vision, Nature Communications, 2020, 11, 6279.

Materials and Methods

Line 63: the authors have to define the acronyms “SBR” and “CMC” the first time they are used.

Reply: We would like to thank the reviewer for the comment. As you suggested, the definitions of “SBR” and “CMC” have been added to the main text in line 64-65.

Line 68: the authors have to insert the unit of relative pressure.

Reply: We would like to thank the reviewer for the comment. The relative pressure here is P/P0, i.e., the equilibrium pressure (P) divided by the saturation pressure (P0); hence, there is no unit for the relative pressure. Nevertheless, we are now providing the range of relative pressure for the BET measurements in the main text (line 69).

Line 104: the authors have to clarify the meaning of negative voltage. 

Reply: We would like to thank the reviewer for the comment. The meaning of the negative voltage has been added to the line 109.

Comment 3: Results and discussion

Figure 3: The sizes of graphics have to increase for improving the reader understanding.

Reply: We would like to thank the reviewer for the comment. As suggested, the size of Figure 3 has now been increased.